# Self-Improving Semantic Perception for Indoor Localisation

**Hermann Blum**
ASL, ETH Zürich
blumh@ethz.ch

**Framcesco Milano**
ASL, ETH Zürich
fmilano@ethz.ch

**René Zurbrügg**
ASL, ETH Zürich
zrene@ethz.ch

**Roland Siegwart**
ASL, ETH Zürich
rsiegwart@ethz.ch

**Cesar Cadena**
ASL, ETH Zürich
cesarc@ethz.ch

**Abel Gawel**
ASL, ETH Zürich
gawela@ethz.ch

**Abstract:** We propose a novel robotic system that can improve its perception during deployment. Contrary to the established approach of learning semantics from large datasets and deploying fixed models, we propose a framework in which semantic models are continuously updated on the robot to adapt to the deployment environments. By combining continual learning with self-supervision, our robotic system learns online during deployment without external supervision. We conduct real-world experiments with robots localising in 3D floorplans. Our experiments show how the robot's semantic perception improves during deployment and how this translates into improved localisation, even across drastically different environments. We further study the risk of catastrophic forgetting that such a continuous learning setting poses. We find memory replay an effective measure to reduce forgetting and show how the robotic system can improve even when switching between different environments. On average, our system improves by 60% in segmentation and 10% in localisation accuracy compared to deployment of a fixed model, and it maintains this improvement while adapting to further environments.

**Keywords:** continual learning, self-supervised learning, online learning

## 1 Introduction

Learning-based systems enable robots to partially understand the environment through, e.g., object detection or semantic classification [1, 2]. Such understanding is a key requirement to enable many complex robotic applications such as autonomous driving or mobile manipulation [3, 4]. However, mobile robots are deployed in increasingly unstructured environments, where learning-based systems often fail [5, 6]. Anticipation of a wide variety of environmental conditions is required for safe operation, however difficult if not impossible. Thus, robotic actors with a high degree of autonomy are required to adapt to unexpected and changing conditions for robust operation. Yet, deployment of learning-based systems typically means training a model on a variety of data and then using this fixed model during deployment. Since it is unattainable for fixed datasets to model all environmental conditions and objects, robustness of current approaches to novel situations is severely limited.

In this work, we explore how learning can be used as a means to self-improve semantic perception during exploration of the environment. Enabling robots to adapt on-the-job poses three main challenges: (i) Models need to be efficiently (re)trained to incorporate new data (*incremental / continual learning*). (ii) Acquired knowledge should be kept while adapting to new tasks and environments (avert *forgetting*). (iii) Training signals are required during deployment, i.e. automatically harvested without manual human supervision in the loop (*self-supervision*). For the first two points, we build upon recent advancements in the field of *continual learning (CL)* [7, 8], also referred to as *incremental learning* [9, 10], and *lifelong learning* [11, 12, 13]. This learning paradigm deals with settings where tasks or classes are presented incrementally, or in which the data distribution changes over time [7].

Key to our proposed system is the combination of CL and self-supervision. In the literature, CL is usually evaluated by consecutively presenting a model with parts of existing datasets. However,

5th Conference on Robot Learning (CoRL 2021), London, UK.

in order to continually learn during deployment, robots require streams of training signals without humans in the loop. We make use of *self-supervised pseudo labels*, which can be generated from multi-sensor [14] or multi-task systems [15]. Our results show that even noisy, imperfect pseudo labels generate useful training signals and – what to our knowledge has not been discovered before – also work well with CL methods that were developed for perfect labels.

To validate our approach, we conduct experiments on a real physical system. We deploy robots in diverse environments with existing 3D floorplans, in which the robot is required to localize. Floorplans are ubiquitous and therefore enable easy deployment into any indoor environment, without the need to generate a map before or during the mission. In applications like construction robotics, the robot's tasks are also defined in these plans. However, robot localisation in floorplans is challenging. While the plan only represents static building structure (*background*), the actual environments contain large amounts of un-modelled objects and clutter (*foreground*), potentially obstructing localization performance [16]. With our system, the robot adapts to each environment by learning to tell apart *background* from *foreground* and improving its localisation in all environments. We further observe catastrophic forgetting when the robot switches between environments, and we identify CL methods that mitigate such forgetting. Finally, our experiments show positive knowledge transfer where the robot improves in novel environments based upon learning in earlier environments. In summary, our contributions are:
• A framework for self-improving perception;
• A pseudo label generation method for background-foreground-segmentation;
• Validation on a real-world robotic system;
• Evaluation of a range of methods to prevent forgetting;
• Evidence that CL with noisy pseudo labels yields significant improvements over fixed models.

All experimental data and code is available at `https://github.com/ethz-asl/background_foreground_segmentation` and a summary video can be accessed at `https://youtu.be/awsynhkkFpk`.

## 2  Related Work

**Self-improving robotic agents** is a both old and underexplored idea. One framework in which agents are self-improving is reinforcement learning (RL). With RL, robots have been learning to walk [17], grasp objects [18], or fly [19]. All these systems indeed learn by self-improving over time, often failing in the beginning of the learning process. Usually these learned models are fixed once they acquire the necessary skills, unlike life-long learning. This is because they require supervision signals, e.g. from simulators that are not available during deployment. However, online adaptation of model-based RL has been shown for example in [20].
Self-improving robotic systems have also been described outside of RL. For example, [21] and [22] describe online parameter optimisations for model predictive control. The adaptive stereo vision of Tonioni et al. [23] has been a particular inspiration for this work. Very related is also Sofman et al. [24], who learn a probabilistic model for terrain traversability in an online and self-supervised fashion. [25] is a parallel work that uses self-supervised learning to improve localisation in a robotic map. Interestingly, the mentioned previous works often do not explicitly address the problem of forgetting, which is a more prevalent problem in our semantic domain adaptation.

**In continual learning**, models are trained from non-stationary data distributions over a series of different tasks [8]. The main objective of CL is to optimize for the performance on each task or domain with which the network is presented at any given time, while achieving positive knowledge transfer between the tasks, and preventing performance on the previous tasks from decreasing. This decrease on previous tasks is commonly referred to as *catastrophic* forgetting [26, 7]. Multiple techniques have been proposed to mitigate such forgetting, ranging from architectural modifications [27, 28] to regularization techniques [29, 30] and methods based on memories [31, 32, 33, 34] or generative models [35, 36]. A number of works have explored the use of CL techniques in the context of semantic segmentation [37, 38, 39, 40, 41]. They however experiment on a class-incremental setting, whereas we are interested in the scenario in which the deployment environment of the agent changes. In this sense, the problem we aim at tackling is also related to a line of works that focus on domain adaptation [42, 43, 44]. However, these works generally assume that both the source and target domain are known at training time, while we tackle the problem of adapting to any (unknown) environment on the fly.

**Self-supervision** is often used to learn useful image features in convolutional neural networks (CNNs). These techniques include learning to (re)color images [45], to (un)rotate images [46], or to relatively position random crops [47]. However, these features require further (supervised) processing to relate them to e.g. classes. In mobile agents, egomotion was found to be a promising self-supervision signal for a range of tasks [48]. Photometric consistency between video frames can be used to jointly learn camera calibration, visual odometry, depth estimation, and optical flow [15].

A different line of works produces pseudolabels for segmentation by leveraging models trained on more available data. Class activation maps of image classifiers [49] can be used to identify object regions as a segmentation proxy [50, 51, 52]. Furthermore, segmentation predictions can be refined by optimizing over Mask R-CNN predictions, room layout estimation, and superpixels [53], by tracking and optical flow [54], or by aggregating predictions in 3D space and projecting them back onto images [14]. While there is no direct prior work for background-foreground-segmentation, our proposed method builds up on similar ideas to use observable characteristics of the environment to produce a learning signal for the target task.

# 3 Proposed System

We propose a self-improving perception system that interlinks localisation within a map and semantic segmentation of the scene. We define the semantics of the scene not as arbitrary class labels, but by the observable affordance that some parts of the scene are mapped (*background*) and some are not (*foreground*). Therefore, we create pseudo labels based on the localisation in the map to train the semantic segmentation, and we use the segmentation into *foreground* and *background* to inform the localisation. This creates a feedback loop that can yield improvements in both parts, as can be seen in Figure 1.

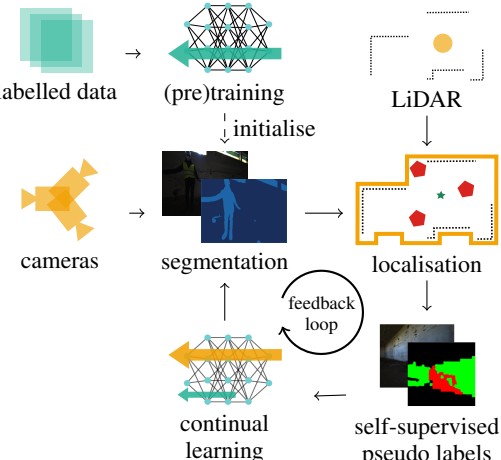

## 3.1 Semantically Informed Localisation

We localize the robot based on aligning 3D Li-DAR scans with the given floorplan in the form of a 3D mesh as in [16]. Given the floorplan mesh $M$, a pointcloud of the LiDAR scan $P$, and an initial alignment $T_{\text{mesh} \to \text{lidar}}^{(t=0)}$, we find subsequent robot poses as

$$T_{\text{mesh} \to \text{lidar}}^{(t)} = \text{ICP}(M, P^{(t)}, T_{\text{mesh} \to \text{lidar}}^{(t-1)}).$$

Figure 1: The proposed self-improving system. The segmentation still incorporates existing training data as a prior, but is not fixed during deployment, as would be the established approach. Instead, our segmentation is updated during deployment based on CL methods. The continual learning uses self-supervised pseudo labels, which are available during deployment without manual labelling. We mark signals from the deployment environment in orange and from the pretraining domain in green.

We use point-to-plane ICP [55][1] and filter out points of large distance and other criteria. Complete parameters are reported in Appendix A.5.

To further divide the scan $P$ into *foreground* and *background* points, we use additional information from a camera system mounted on top of the LiDAR. Once camera images are semantically segmented, we filter $P_{\text{background}} \subseteq P$ as those points $p \in P$ whose reprojected pixel in image frame is segmented as *background* and localise with $\text{ICP}(M, P_{\text{background}}^{(t)}, T_{\text{mesh} \to \text{lidar}}^{(t-1)})$.

## 3.2 Pseudolabel Generation

We generate pseudo labels for each camera based on the LiDAR pointcloud that is localised w.r.t. the floorplan. First, for each point of the localised LiDAR scan, we calculate the distance to the closest plane of the mesh using fast intersection and distance computation [57]. We then check if the distance surpasses a given threshold $\delta$. If so, the point is assigned the *foreground* class, otherwise the

---

[1]The proposed framework does not require a specific registration method. For registrations with large overlap, point-to-plane ICP was found a sufficient solution in [56] and subsequently used in similar settings [16].

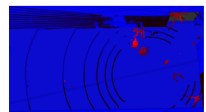 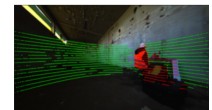 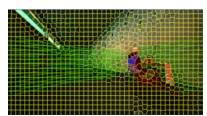 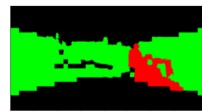

(a) Mesh and Pointcloud    (b) Projected Pointcloud    (c) Oversegmented Image    (d) Pseudo Labels

Figure 2: Generation of pseudo labels. After localising the LiDAR scan, points are labeled by comparing the distance to the floorplan with a threshold $\delta$ (red is foreground, green is background). They are then projected into the image and aggregated into superpixels.

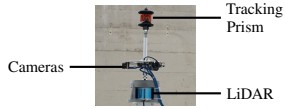 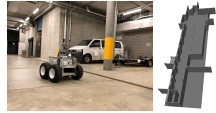 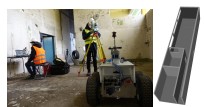 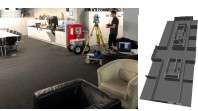

(a) Multi-Sensor Setup    (b) Parking Garage    (c) Construction Site    (d) Office Environment

Figure 3: Overview of our experimental setup.

*background* class. In the second stage, we project each point onto the respective camera frames and refine the projection using SLIC superpixels [58]. In particular, we first oversegment the image into a superpixel set $S$. A superpixel $s \in S$ is then assigned a class according to a majority voting of the contained projected labels. We further improve the segmentation by discarding superpixels whose depth variance surpasses a given threshold. An overview of the approach is depicted in Figure 2.

These pseudo labels are not optimal[2]. The goal however is not to generate perfect labels, but a useful training signal that can be generated on-the-fly without requiring any external supervision. As our experiments demonstrate, even this noisy learning signal substantially boosts performance.

### 3.3 Domain Adaptation with Continual Learning

To solve the task of background-foreground segmentation, we incrementally train a neural network architecture on different data sources. We pre-train a CNN on the NYU-Depth v2 dataset [59], which contains 1449 images extracted from video sequences of indoor scenes, each with per-pixel semantic annotations. We map the classes *wall*, *ceiling*, and *floor* to background and regard everything else as foreground. This (pre-)training step allows the model to acquire prior knowledge as an inductive bias to perform the same segmentation task on subsequent environments that the agent is presented with.

During deployment, the network is then updated with self-supervision through the pseudolabels generated on the current scene. With reference to the nomenclature often used in CL [8], we consider each new environment a *task*, and we assume *task boundaries* to be known, since for each task, the robot is given a different floorplan. Every time the robot is moved to a new environment, the same scheme is applied, i.e., the network trained on the previous environments is updated by training on the pseudolabels from the current environment.

To prevent forgetting of information learned on the previous tasks, we adopt a method based on memory replay buffers. When adapting to a new environment, each training batch is filled with the frames collected in the current environment, along with a small fraction of images collected in the previous environments. Therefore, the model jointly optimizes over current and previous environments. However, storing all observations from past environments in memory would come at huge costs. Instead, a memory buffer for each previous environment only contains a random subset of all images and (pseudo-)labels. Details of the buffer implementation and a comparison with other CL frameworks are described in Section 4.4.

---

**Algorithm 1:** Memory Replay

memory_buffer = RandomChoice(
   10% out of previous_scene)
train_data = ShuffleTogether(
   current_scene, memory_buffer)
**while** *training not converged* **do**
   **foreach** *batch in train_data* **do**
     randomly augment batch
     train on batch
   **end**
   validate on 10% hold-out of
     current_scene
**end**

---

## 4 Experimental Evaluation

We test and verify the applicability of the proposed framework in different steps of increasing complexity. We first validate that our robot can self-improve by deploying it into different unknown

---

[2]For example, for any clutter object standing on the ground, all points below a height of $\delta$ may be labelled as *background* in the pseudolabels, dependent on the boundaries of the superpixel.

| environment | mean/median/std translation error [mm] | | | segmentation quality [% mIoU] | |
|---|---|---|---|---|---|
| | no segmentation | trained on NYU | self-improving | trained on NYU | self-improving |
| Garage | 50 / 41 / 37 | 58 / 41 / 73 | **43 / 35 / 31** | 33.9 | **62.8** |
| Construction | 488 / 183 / 999* | 126 / 78 / 129 | **104 / 68 / 105** | 27.6 | **48.2** |
| Office | 167 / 168 / 88[+] | 196 / 145 / 202 | **150 / 138 / 81** | 46.5 | **53.9** |

Table 1: Deployment to a single novel environment. We observe that segmentation in general is advantageous and the CL on self-supervised pseudolabels improves in all environments compared to deploying a fixed model. * marks ICP failures. [+] uses adapted parameters, see Appendix A.5.

environments and measure the gained improvement. We then evaluate the effects of forgetting and knowledge transfer when switching deployment between different environments. Finally, we conduct an experiment in which the robot learns online during the mission. For each experiment, we measure the localisation error in the x-y plane[3] in mean, median and standard deviation. We also measure the segmentation quality in mean Intersection over Union (mIoU).

**Robotic Platform.** We conduct all our experiments on a wheeled robotic ground vehicle [60] with the sensor setup shown in Figure 3a. We calibrate all cameras to an IMU that we attach to the sensor system and then align trajectories from visual-inertial odometry and LiDAR to find the extrinsic calibration between the LiDAR and the cameras. The tracking prism is used to gather ground-truth with a total station for our evaluation and is mounted on the sensor system to be aligned with the optical center of the LiDAR. For time synchronisation, we trigger all cameras simultaneously. From images captured at 20 Hz we then take those closest to the timestamp of the LiDAR scans, which are captured at 5Hz. Between the external total station and the robot, we correct time-offsets manually.

**Evaluation Environments.** We deploy our proposed system into three different environments: a construction site, a parking garage, and an office floor. Architectural researchers provided us 3D meshes, constructed from dense 3D scans (Construction) and existing 2D floorplans supplemented with additional measurements (Garage and Office). Figure 3 shows these meshes together with the experimental setup. In each environment, we steer the robot through multiple independent trajectories of 2-3 min while tracking the robot with a total station. The coordinate system of the total station is always initialised to the origin of the building mesh, which we therefore set at corners visible to the total station. To evaluate the learned segmentation models, we sample and annotate around 30 ground-truth segmentations per environment. We then evaluate the segmentation predictions against ground-truth annotations within a static field-of-view (FoV) mask of the LiDAR[4].

**Learning Setup.** We use a lightweight architecture based on Fast-SCNN [61] that we train for background-foreground segmentation. We resize input images to a common size ($480 \times 640$ pixels). We first train the model on the NYU dataset. Then, when we deploy the robot into a new environment (cf. Sec. 4.1), we fill a replay buffer with samples from NYU in addition to training on the pseudolabels from the current environment. When we evaluate the transfer from a first environment to a second one (cf. Sec. 4.2), we replay images from both NYU and the first environment. We set the replay fraction to $10\%$, but evaluate different replay regimes and strategies in Section 4.4. Before feeding images to the network, we augment them with left-right flipping and random perturbation of brightness and hue. In each experiment, we hold out $10\%$ of the training samples for validation and train our model using Adam optimizer. See Appendix A.2 for all training parameters and Appendix A.1 for runtimes.

## 4.1 Deployment in a new environment

To test the effectiveness of the pseudolabel training and the localisation based on filtered pointclouds, we deploy the robot in a first new environment. There, the robot collects information in the form of pseudolabels, which are used to train the segmentation. Afterwards, we deploy the robot over a different trajectory in the same environment and measure the performance of both segmentation and localisation.

In Table 1 we compare the performance obtained from our self-supervised foreground-background segmentation ('self-improving') with segmentation obtained from the network pre-trained on NYU

---

[3]The ground truth from the total station only provides translation measurements and does not enable evaluation of correctly estimated orientation.

[4]The FoV mask has two reasons: Because we are primarily interested in good localisation, comparing segmentation quality in the region that can be reprojected to the LiDAR relates better to localisation performance. Additionally, pseudolabels are also only available in image regions where the LiDAR provides information, therefore we expect the segmentation to learn mostly the semantics of the scene visible in these regions.

| environment sequence NYU → A → B (→ C) | method | mean/median/std translation error [mm] A | B | C | segmentation [% mIoU] A | B | C |
|---|---|---|---|---|---|---|---|
| Garage→ Construction | replay | 41 / 33 / 30 | 98 / 66 / 98 | | **60.8** | 48.6 | |
| | finetuning | 40 / 33 / 29 | 87 / 67 / 77 | | 55.1 | 49.4 | |
| Garage→ Office | replay | 39 / 31 / 31 | 168 / 137 / 109 | | **62.6** | 47.2 | |
| | finetuning | 37 / 30 / 33 | 196 / 118 / 267 | | 61.0 | 47.4 | |
| Construction→ Garage | replay | **105 / 68 / 108** | 40 / 33 / 29 | | **49.3** | 62.2 | |
| | finetuning | 549 / 84 / 1500* | 40 / 31 / 29 | | 42.3 | 62.0 | |
| Construction→ Office | replay | **125 / 72 / 128** | 158 / 137 / 93 | | **50.3** | 47.6 | |
| | finetuning | 514 / 191 / 914* | 146 / 123 / 93 | | 45.4 | 49.4 | |
| Office → Garage | replay | 153 / **131** / 95 | 44 / 34 / 36 | | **47.8** | 62.1 | |
| | finetuning | 182 / 151 / 114 | 38 / 32 / 27 | | 40.2 | 61.0 | |
| Office → Construction | replay | 171 / 161 / **86** | 91 / 66 / 85 | | **47.5** | 49.9 | |
| | finetuning | **168 / 159 / 88** | 121 / 70 / 128 | | 33.3 | 49.1 | |
| Construction → Garage → Office | replay | **105 / 72 / 92** | 39 / 30 / 29 | 167 / 129 / 113 | **50.4** | **64.7** | 45.2 |
| | finetuning | 385 / 95 / 868* | 41 / 32 / 32 | 145 / 114 / 130 | 37.2 | 62.2 | 46.3 |
| Office → Garage → Construction | replay | 157 / **132** / 102 | 41 / 31 / 32 | 105 / 70 / 100 | **43.9** | **63.2** | 50.3 |
| | finetuning | 158 / 145 / **85** | 43 / 35 / 30 | 112 / 82 / 92 | 33.8 | 57.2 | 48.2 |

Table 2: Evaluation of forgetting and knowledge transfer when switching between deployment environments. The perception system is subsequently on pseudolabels of the environments in the given order and then evaluated on all. Bold marks cases where one method reduces forgetting compared to the other method. Underlined metrics are better than single-environment deployment from Table 1. Finetuning leads to three cases (marked with star) where forgetting causes ICP failures.

('trained on NYU'), and with a baseline that does not semantically filter the pointcloud ('no segmentation'). We note that segmentation in general is important for good localisation and can prevent failure, as on the construction site. The results also confirm the effectiveness of the pseudolabels, as training on these yields improvements in both segmentation quality and localisation error in all metrics. We conclude that the self-improving setup is working as expected and that through the feedback loop indeed the localisation improves the segmentation, and the segmentation improves the localisation.

## 4.2 Transfer into multiple environments

In a second stage of experiments, we evaluate the ability of our system to retain knowledge and still adapt to new domains when moved from a first environment to a second (and third) one. In particular, we iteratively train the same model on pseudolabels from a series of environments. We then evaluate the extent of forgetting on both the pretraining data (NYU) and any environment before the last one, comparing our adopted method based on a replay buffer with simple fine-tuning.

As shown in Table 2, using replay buffers improves the segmentation performance on the past tasks w.r.t. the case in which no replay is adopted. This indicates successful mitigation of forgetting and is even more prominent when measured on the pseudolabels and measuring forgetting on the NYU data, which we analyse in more detail in Table 5 in Appendix A.3. Table 2 further shows that the forgetting of finetuning can cause localisation failure on the source environment. Replay prevents such failure successfully. In the CL literature, memory replay is usually studied with high-quality segmentation masks covering all pixels in the image [7]. Yet, from our observations we can conclude that memory replay is also effective when the replayed labels are noisy and imperfect.

We note that finetuning can result in better adaptation to the target environment, especially with regard to localisation. This is expected, since the learning process of finetuning is fully tailored to the new environment. This effect is known as *stability-plasticity dilemma* [62], i.e. old knowledge can inhibit learning of new knowledge, but increasing plasticity can in turn lead to forgetting. In our experiments, the relative improvements of finetuning over memory replay that we observe on the target environment are marginal, suggesting that the chosen 10% replay finds a good balance.

Table 2 also highlights cases in which deployment in multiple environments is better than improving only towards one specific environment (Section 4.1), both with and without memory replay. This indicates positive knowledge transfer between our evaluation environments. For a self-improving robotic system this is a promising finding, showing that the robot not only adapts to the target environment, but generally improves at its task with every new deployment.

Finally, we observe that segmentation quality and localisation error are sometimes inconsistent, where a drop in segmentation quality in a past environment does not necessarily transfer into worse localisation. This indicates potential for advanced methods which only keep in memory those observations that are important for the task at hand.

## 4.3 Online Learning

The previous experiments have been conducted in a multi-mission fashion, in which a first mission gathers data of the environment and the robot learns from it to improve in subsequent missions. Figure 4 now shows a case study how the system can learn online during a mission by learning from the pseudolabels directly as they are generated. We initialise the model pretrained on NYU and observe that over time segmentation quality increases and localisation error decreases. Notably, already a few seconds of online learning increase the segmentation quality significantly. However, it takes longer until we measure a notable effect on the localisation. Due to the limited time until the end of the trajectory, we are therefore unable to see if there is a feedback effect where the improved localisation would create better pseudolabels that can in turn increase the segmentation quality further. For this, further investigation on longer deployments is necessary. We present similar studies for the other environments in Appendix A.7.

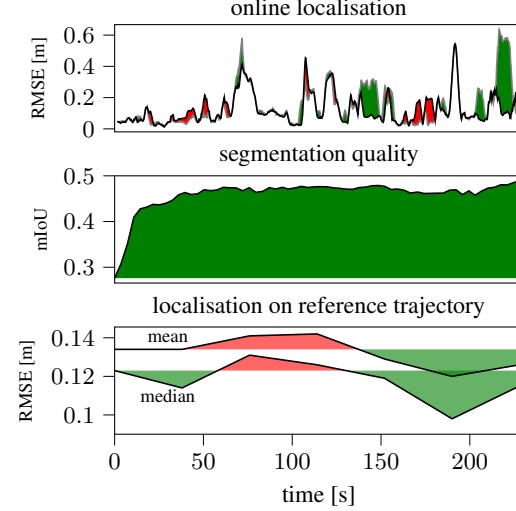

Figure 4: Online Learning on the construction site. We measure how the robot improves online (row 1), but also evaluate snapshots of the model on reference data (rows 2 and 3). Areas in green/red show changes (better/worse) with respect to a fixed model. We observe that segmentation quality increases over time and localisation error decreases.

## 4.4 Ablation Studies

Since CL methods have not been explored in a real-world robotic context, and even less in combination with noisy pseudolabels as training signal, we perform an extensive ablation of different methods and parameters to validate our use of replay buffers.

We first evaluate two different strategies for memory replay. In the first strategy, which is the one that we adopt in the main experiments, on each source-to-target experiment (e.g., NYU→Garage), we fill the replay buffer with a fraction of samples from the source dataset(s) (NYU in the example), which we select randomly. We then fill training batches from the replay buffer and target dataset according to their relative sizes. In the second strategy, we fill the replay buffer with the full source dataset but fill training batches with a pre-defined target-source ratio. For instance, a ratio Garage : NYU = 4 : 1 with a batch size of 10 indicates that batches on average contain 8 images from Garage and 2 images from NYU. As shown in Table 3, milder replay regimes (larger target-source ratios, or smaller replay fractions) achieve higher performance on the target domain, but cause the amount of information retained from the source domain to drop. This forgetting phenomenon is particularly evident in the adaptation tasks in which the semantic gap between source and domain is larger. Indeed, for instance, in the NYU→Garage experiment we observe a drop of 31.9% in mIoU on the NYU labels between a replay strategy with a fraction of 10% and simple fine-tuning, while the same decrease in performance when the target domain is Office – semantically closer to the indoor dataset NYU – is 14.8%. At the same time, the segmentation quality on the pseudo-labels of the target dataset follows an inverse trend, generally increasing for smaller amounts of replay. This highlights the trade-off between the accuracy on the target and on the source domain.

Memory replay comes at the cost of storing images and labels of past environments, which does not scale well to more than a few environments. We therefore further compare regularization techniques from the continual-learning literature, which try to prevent forgetting without storing as much data. In particular, we investigate the use of a distillation approach, in which a regularization term is added to the cross-entropy loss, to encourage the network to retain the knowledge from previous tasks. Similarly to [37], we apply this distillation either on the output logits produced by the network (*Output distillation*) or on the intermediate features (*Feature distillation*). Furthermore, we evaluate Elastic Weight Consolidation (EWC) [29]. We present further details in Appendix A.4.

As shown in Table 3, in our experiments replay buffers prove to be the most effective among the examined methods in minimizing the amount of forgetting on the NYU dataset, and generally allow attaining a good trade-off with the segmentation quality on the pseudo-labels from the target domain.

| | | NYU→Garage | | | NYU→Construction | | | NYU→Office | | |
|---|---|---|---|---|---|---|---|---|---|---|
| | | NYU | Garage | | NYU | Construction | | NYU | Office | |
| | | | Pseudo | GT | | Pseudo | GT | | Pseudo | GT |
| Finetuning | | 36.4 | 96.3 | 61.8 | 36.6 | 79.5 | 48.9 | 66.2 | 70.9 | 51.2 |
| Replay buffer with ratio target : NYU | 1 : 1 | **87.2** | 86.5 | 61.5 | **82.9** | 66.6 | 46.1 | **83.5** | 57.8 | 49.9 |
| | 3 : 1 | 81.1 | 91.7 | 60.7 | 79.8 | 73.1 | 47.8 | 81.4 | 68.1 | 52.2 |
| | 4 : 1 | 79.8 | 92.4 | 62.3 | 78.7 | 73.1 | 48.8 | 82.0 | 65.8 | 51.7 |
| | 10 : 1 | 73.4 | 94.7 | 61.3 | 75.3 | 75.6 | 47.6 | 77.7 | 71.2 | 52.1 |
| | 20 : 1 | 67.5 | 95.3 | 62.0 | 72.4 | 76.3 | 48.3 | 76.0 | 69.1 | 52.0 |
| | 200 : 1 | 53.9 | 96.1 | 61.6 | 53.2 | 77.2 | 48.7 | 74.8 | 68.7 | 50.9 |
| Replay buffer with fraction replay | 10% | **68.3** | 95.4 | 62.8 | **78.6** | 77.0 | 48.2 | **81.0** | 69.7 | 53.9 |
| | 5% | 65.0 | 95.9 | 62.0 | 76.2 | 76.9 | 48.5 | 79.9 | 69.3 | 51.5 |
| Feature distillation | $\lambda = 0.5$ | 35.4 | 96.1 | 61.9 | 33.2 | 77.1 | 50.3 | **65.3** | 71.1 | 50.7 |
| | $\lambda = 1$ | 34.8 | 95.9 | 61.2 | 30.6 | 76.9 | 50.8 | 58.6 | 78.9 | 49.1 |
| | $\lambda = 10$ | **37.4** | 94.2 | 61.6 | **33.6** | 72.1 | 48.3 | 48.7 | 72.2 | 48.0 |
| | $\lambda = 50$ | 33.6 | 92.7 | 61.1 | 32.6 | 63.1 | 46.5 | 44.0 | 64.5 | 46.8 |
| Output distillation | $\lambda = 0.5$ | 33.1 | 94.4 | 63.3 | 34.0 | 76.5 | 47.8 | **62.9** | 68.4 | 49.9 |
| | $\lambda = 1$ | 32.4 | 85.3 | 64.4 | **38.2** | 59.4 | 46.6 | 53.0 | 60.4 | 44.3 |
| | $\lambda = 10$ | 37.8 | 40.8 | 47.5 | 37.9 | 32.9 | 37.1 | 45.3 | 35.8 | 36.1 |
| | $\lambda = 50$ | **39.0** | 48.9 | 53.0 | 31.7 | 28.7 | 31.1 | 46.3 | 30.5 | 35.9 |
| EWC [29] | $\lambda = 0.5$ | 36.1 | 96.4 | 61.5 | 34.4 | 76.5 | 47.9 | 65.7 | 69.2 | 51.6 |
| | $\lambda = 1$ | 36.2 | 96.3 | 61.4 | **37.0** | 76.4 | 48.0 | 66.2 | 74.0 | 50.8 |
| | $\lambda = 10$ | **37.9** | 96.2 | 61.1 | 35.4 | 76.2 | 48.0 | **70.6** | 69.1 | 51.8 |
| | $\lambda = 50$ | **37.9** | 95.9 | 61.5 | 35.0 | 75.6 | 47.9 | 65.5 | 73.2 | 51.0 |

(Right-side brackets group "Replay buffer with ratio" and "Replay buffer with fraction replay" under *memory replay*; "Feature distillation", "Output distillation", and "EWC" under *regularisation methods*.)

Table 3: Ablation study over different CL methods. After adapting from the source domain (NYU) to the different deployment environments, we measure segmentation quality [% mIoU] on the source data as well as the self-supervised pseudolabels (Pseudo) and our ground-truth annotations (GT). Forgetting is therefore indicated by low performance in the gray columns. We compare the finetuning with memory replay and regularisation methods. We observe that memory replay is more successful than regularisation at preventing catastrophic forgetting in our application.

We also note that, with limited exceptions, both regularization approaches fail to maintain a good performance on the source dataset NYU. We believe that for distillation methods this can be ascribed to the fact that, as opposed to related works that explored similar techniques in a class-incremental setting [37, 38], we conduct domain adaptation. More importantly, unlike [37, 38] our supervision signal does not consist of accurate ground-truth annotations available at all pixels, but of noisy pseudo-labels that often cover a limited region of the image. Finally, while similar considerations also apply for EWC, we believe that the limited effectiveness of this technique in our setting is also related to the method being designed for classification as opposed to image segmentation.

## 4.5 Limitations

Our system required some manual intervention in the sense that a few parameters were changed across environments (ICP and superpixels, see Appendix). While coming close, the system also did not run in real-time. We are confident that both points can be tackled with better software implementations.

This study is limited to binary segmentation. However, our framework itself is agnostic of the self-supervision method and can be directly extended to more classes based on multi-sensor [14] or multi-task systems [15], or temporal consistency [54]. More affordances can be observed through manipulating objects [63], temporal change [25], or context and spatial co-occurrence [64, 65].

## 5  Conclusion & Outlook

In this work we propose a framework for self-improving perception by combining CL with self-supervision. We study this on a real robotic system that localises in 3D floorplans. Our experiments validate the gains of the self-improving systems in diverse environments. In particular, we analyse the effects of knowledge transfer and forgetting when switching between environments. We find that memory replay is an effective solution that can mitigate forgetting, and observe that the system can even further improve as it transfers knowledge from previously seen, distinct environments.

The main finding from our evaluations of knowledge transfer and forgetting is that self-supervision and CL is an effective combination, even if the self-supervision is noisy. This opens up exciting questions for future research. Through the self-supervision approaches that we describe in Sections 2 and 4.5, the same framework can be transferred to other perception tasks such as more semantic classes or perception for manipulation. For some tasks, task boundaries may be less clear, and further research is necessary to lift this assumption. Finally, we identify long-term dynamics and stability of self-improving systems as an interesting direction for future research.

**Acknowledgments**

This work was partially funded by the Hilti Group. Eberhard Unternehmungen kindly allowed us to conduct experiments on their premises.

Furthermore, we thank Selen Ercan for creating the building models, Florian Tschopp and his VersaVIS board for all the multi-sensor calibration, Shen Kaiyue for initially implementing and testing various regularisation based continual learning methods, and Andrei Cramariuc for GPU cluster support.

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
