# OpenReview forum: "Self-Improving Semantic Perception for Indoor Localisation"
_robot-learning.org/CoRL/2021/Conference — CoRL2021 Poster_

### Official Review · Reviewer_Ncrf · 2021-07-12

**Originality:** Very Good
**Technical Quality:** Very Good
**Clarity Of Presentation:** Excellent
**Impact:** 4

**Recommendation:**

Strong Accept: I recommend accepting the paper and will argue for my recommendation even if other reviewers hold a different opinion.

**Summary:**

The paper combines self-supervision with methods from continual learning to enable a robot to improve its localization during deployment. More concretely, the proposed method automatically labels images that are collected during deployment based on localization information and uses them to continuously update a segmentation module, which in turn is used for localization. The authors show in a suite of real-world environments that this feedback loop between segmentation and localization modules significantly improves both the localization capabilities and the quality of generated segmentations. They also note that updating the segmentation module in an online fashion comes with the risks of catastrophic forgetting of previously learned features, and show that a simple memory buffer can prevent this from happening.

**Issues:**

- Figure 1 does not really seem to add much to the paper. I would much rather see a qualitative comparison, e.g., in the form of a suitable frame from the summary video in the supplement.
- The images in Figure 4 have pretty high resolution, which can lead to some lag when loading the page in a .pdf reader. Down-sampling them would help, and also decrease the overall size of the file.
- In the "Robotic Platform" subsection of Section 4, it says that images are captured at 20Hz and then matched to the 5Hz LiDAR scans, with offsets being corrected by hand. Is there a reason for the images to be captured at 20Hz instead of using 5Hz as the common denominator?
- The section on online learning (Section 4.3) is rather short. While the paper states that further investigation on this topic is needed, it would still be nice to present the existing results more clearly. For example, Figure 5 Row 3 shows that the localization results get worse for some time when online learning, before eventually improving upon the baseline. It would be interesting so see why this is the case. Similarly, Figure 5 Row 1 shows relative changes to a baseline. Adding some general trend, e.g., an exponential moving average or cumulative histogram of the changes, could help readability. Finally, it would be nice to see how statistically significant these changes are, though this may be out of scope for the revision.

### Misc.

- Line 43 "... has not been discovered before" seems like a rather strong statement. Adding some variant of "to the best of our knowledge" would alleviate this. A similar argument can be made in Line 277.
- In Table 3, the "61.5" in the last row is written in bold, which seems to be unintentional.

**Reviewer Expertise:**

Good: General knowledge of the area

**Strengths And Weaknesses:**

### Strengths

- The main idea of enabling continual learning for robot localization via self-supervised segmentation labels is both interesting and well-executed.
- The paper is presented in a clear and concise fashion.
- The approach is applied to different real-world environments, showcasing its versatility and making for a convincing experimental section.
- A series of ablation experiments is conducted to show the influence of different hyperparameters and design decisions on the overall framework. Different methods of continual learning are explored and compared to the memory buffer chosen for the final model.

### Weaknesses

- The automatic segmentation labeling requires a ground truth floor-plan that is compared to LiDAR scans. This limits the applicability of the approach to indoor environments with precisely known layouts.
- While the use of a sample buffer helps prevent catastrophic forgetting, the ablations show that it comes with the cost of reduced performance on new environments. As such, the question of how many old samples to use becomes a hyperparameter that needs to be carefully tuned for a downstream application. Note however that this is well-addressed in the ablation studies.
- As stated in the "Limitations" section, some parameters need to be tuned depending on the environment. This raises the question of how well the method generalizes to different and potentially more complex environments.

**Summary Of Recommendation:**

The paper explores the idea of improving semantic segmentation during robot deployment by means of continual learning and self supervision. This is an interesting contribution on its own, and naturally opens up various directions for future work.

Presentation-wise, the main ideas of the paper are thoroughly motivated and explained clearly. The experimental section clearly shows the benefit over the approach over a baseline that does not use it, and a number of ablations give insight into the importance of individual components. The main shortcoming of the experiments seems to be the need for manual parameter tuning. Adding some measure of confidence (e.g., standard deviations) would help to make a better case for the approach.

Overall, the paper introduces and experimentally supports an interesting novel approach, and presents everything in a clearly understandable manner. I would thus recommend to accept it.


**Update**
The authors have addressed most of my concerns during the revision period. I am still not entirely satisfied with the scope and presentation of Section 4.3 (as well as the related Appendix 7), and agree with the other reviewers that the paper would benefit greatly from a pseudo-label baseline. However, the pros of the paper far outweigh the few problems I have with it, causing me to keep my score of "Strong Accept".

---

> ### Author Response · Authors · 2021-08-19
> **Answer to Reviewer Ncrf: Part 2 of 2**
>
> > The section on online learning (Section 4.3) is rather short. While the paper states that further investigation on this topic is needed, it would still be nice to present the existing results more clearly. For example, Figure 5 Row 3 shows that the localization results get worse for some time when online learning, before eventually improving upon the baseline. It would be interesting so see why this is the case. Similarly, Figure 5 Row 1 shows relative changes to a baseline. Adding some general trend, e.g., an exponential moving average or cumulative histogram of the changes, could help readability.
>
> We agree on the need for a general trend. This is why we plotted rows 2 and 3, which compare again relative to the baseline (which is marked by the green and red areas), but instead of the current location or a moving window around the current location of the robot, we evaluate the learned model at that time on reference data. We considered this a stronger comparison than a moving average, which is still heavily dependent on the current location of the robot. As can be seen in the supplementary video, the robot moves from a larger space into a smaller space and back to the larger space. A moving average would therefore most of the time only consider the localisation accuracy in one of these spaces. We are however happy to include a moving average filter if the reviewer disagrees with this judgement.
>
> Concerning the interpretation of Figure 5: When evaluating on the reference trajectory, we see this slight degradation in localisation accuracy at 80s. This is when the robot entered the smaller room for some time. It is therefore likely that the overall localisation accuracy decreases because at that moment, the robot has tuned itself more towards the small room, and the big improvements are only noticeable after it explored both spaces sufficiently. We did not add this discussion to the paper because we consider this data not sufficient to draw such conclusions, and a more thorough investigation of which frames to keep in the training data buffer, which learning rate to apply, at what speed to train etc. would be necessary for the online learning, but would by far exceed the page limit for this work. Instead, we added this (small) experiment to demonstrate that this capability exists at all, because we consider it very useful and exciting for future work.

---

> ### Author Response · Authors · 2021-08-19
> **Answer to Reviewer Ncrf: Part 1 of 2**
>
> > The automatic segmentation labeling requires a ground truth floor-plan that is compared to LiDAR scans. This limits the applicability of the approach to indoor environments with precisely known layouts.
>
> Given the challenges that we demonstrate on localising in the floorplan, we would not consider it ‘ground truth’. While localisation against floorplans or 3D models is necessary for most kinds of construction robotics (because most tasks are defined in these models), we still consider it very useful for any other indoor robot. Floorplans are ubiquitous, e.g. most flats and houses are advertised with floorplans, any commercial building must have one to show fire escape routes, etc. There are multiple software solutions available that convert these drawings into 3D meshes like the ones used in our work, e.g. [A, B]. Given also the questions of other reviewers, we clarified this in the introduction. Please let us know if further clarifications should be added.
>
> >  Adding some measure of confidence (e.g., standard deviations) would help to make a better case for the approach.
>
> If we understand this comment correctly, we should perform an ablation of the used ICP parameters to measure how sensitive the method is to their tuning. We propose to do this in the office and construction environment, investigating what happens if we change the parameters between the 2 configurations. We hope to be able to present results in the appendix by next week.
> However, if we misunderstood this comment, please let us know right away.
>
> > Figure 1 does not really seem to add much to the paper. I would much rather see a qualitative comparison, e.g., in the form of a suitable frame from the summary video in the supplement.
>
> Thank you for the suggestion. We tried to find a good summary graphic, but we understand now that what we had was not easy to follow. We removed the figure to have extra space for the revision, and instead link to the summary video in the introduction.
>
> >  Is there a reason for the images to be captured at 20Hz instead of using 5Hz as the common denominator?
>
> The reason is hardware related. Our LiDAR does not have a hardware signal that could be used to trigger the Cameras in a synchronised fashion. Triggering them independently of the LiDAR at 20Hz and subsequently identifying the frames closest to the LiDAR timestamp therefore was the best available option in our opinion. Given that the exposure time of the images is in any case much shorter than the 200ms for each scan, we did not see a need for stronger synchronisation. It is important to note that this alignment is done automatically by synchronising the time between LiDAR, cameras and the robot’s main computer. Manual alignment of time-offsets was only required between the robot and the external total station that generated the ground-truth localisation. We changed the final sentence in Line 189 to clarify this.
>
> [A] https://www.homestratosphere.com/convert-2d-floor-plan-image-to-3d-floor-plan/
> [B] https://www.archilogic.com

---

> > ### Author Response · Authors · 2021-08-30
> > **Ablation Study Completed**
> >
> > We thank the reviewer for the suggestion of testing the dependency between the ICP parameters and the method. In Appendix A.5, we now present an ablation of the full system, ie localisation -> pseudolabel collection -> training -> localisation with respect to those parameters and present results in Table 7. We take pseudolabels from different trajectories as the final measured localisation performance, but use the same ICP parameters throughout one data point.
> > As it turns out, our final system is very robust to these ICP parameters. For most cases, our system improves over the baseline regardless of the parameters. While different parameters for the full localisation in the office were used during our development process, it turns out that they are unnecessary for our final system.
> > At an early stage of development, the default parameters were not allowing us to acquire meaningful pseudolabels in one of the office trajectories (the first one we experimented with), therefore the choice for a different ICP configuration. We are extremely thankful for the reviewers to suggest this ablation with our final system, which shows that our self-improving method is actually working across all environments without specializing the ICP parameters for localisation or obtaining the pseudolabels. Yet, we kept Table 1 and the note in the Limitations section as-is to be consistent with the other experiments, and refer to the Appendix where we then discuss the ablation study.

---

### Official Review · Reviewer_jkw9 · 2021-07-21

**Originality:** Fair
**Technical Quality:** Fair
**Clarity Of Presentation:** Fair
**Impact:** 3

**Recommendation:**

Weak Reject: I recommend rejecting the paper, but will not argue for my recommendation if the majority of other reviewers have a different opinion.

**Summary:**

This paper presents a self-supervised and continual learning framework for improving indoor foreground-background segmentation and robot localization with known maps/models. The self-supervised learning approach involves generating pseudo segmentation labels from LiDAR scans and using them to fine-tune a segmentation model pre-trained with an offline segmentation dataset (NYU-Depth). The paper also extends this to continual learning where the robot acquires pseudo-labels on the fly and fine-tunes the model in new environments. This involves using a memory replay buffer to avoid catastrophic forgetting of previous experiences. Experimental results show that both the self-supervised and continual learning lead to significant error reductions in foreground-background segmentation.

**Issues:**

Major
- See comments above on evaluation.

Minor
- L60: These contributions can be summarized succinctly. For example, the pseudo-label generation can be part of the self-improving framework.


**Reviewer Expertise:**

Good: General knowledge of the area

**Strengths And Weaknesses:**

Strengths:
- In general, self-supervised and continual learning are both quite relevant for real-world robotic systems. While data-driven paradigms have dominated vision methods, in robotics, systems have to rapidly adapt to new environments. And learning could be an interesting solution here.
- The paper presents a real-robot setup with real-world evaluations.

Weaknesses:
- The self-supervised learning approach uses pseudo-labels generated by thresholding LiDAR scans to mask out foreground from background, but there is no baseline that investigates how good these pseudo-labels themselves are. What if you don't learn anything (self-supervised or continual) and directly use these pseudo-labels for segmentation? How well does this perform? The pseudo-labels might be noisy, but this depends on what sort of accuracy is necessary for the end-task, which is undefined. I am not sure if this pseudo-label result is already in Table 1; I had a hard time connecting the descriptions in Section 4.1 to items in Table 1.
- The results emphasize 60% gains in foreground-background segmentation from the proposed approach, but the paper does not mention a strong use-case for why foreground-background is necessary or important for a specific task. The current setup assumes access to LiDAR+vision and maps/models of the environment, so I am not sure what foreground-background segmentation offers in this scenario. Obstacle avoidance and localization can be done with LiDAR, without vision. Further, the segmentation labels do not classify object categories, so they cannot be used for task-planning.
- While the real-world evaluations are really great, they are hard to reproduce. Would it be possible to benchmark the performance gains in a simulated setup? Compared to the previous concerns, this is relatively minor, but nonetheless important. It’s hard to generate photorealistic simulations of construction and parking sites, but equivalent scenes from NVIDIA Isaac Sim or just setting-up a Gazebo environment might be feasible. While the 10% improvements in localization is great, this result can be substantially strengthened if it can be replicated in a reproducible benchmark. Similarly, the experiments in Section 4.3 need to be empirically more rigorous. What happens after 250 secs? Why does the plot stop exactly when the segmentation quality peaks?


**Summary Of Recommendation:**

While the proposed self-supervised and continual learning approaches are interesting, the evaluation is missing an important baseline and a strong motivating use-case for foreground-background segmentation.

**Post Rebuttal**
I thank the authors for the detailed follow-up responses. Without the pseudo-label baseline, I find it hard to appreciate the value of the proposed learning-based methods. "we never claim that different solutions to the application would not be possible" -> I understand that there is merit to working on approaches that have other viable alternatives, but without empirical evaluations against simpler baselines, it's hard for the community to figure out what is the best way to _solve the problem_ (i.e. localization), even if the novel approach might be promising for some future applications. So I am keeping my original score of Weak Reject.

---

> ### Author Response · Authors · 2021-08-19
> **Answer to Reviewer jkw9: Part 2 of 2**
>
> > While the real-world evaluations are really great, they are hard to reproduce. Would it be possible to benchmark the performance gains in a simulated setup?
>
> Please have a look at our code supplement. This will be published alongside the rosbag recordings, the annotated datasets, and our trained networks. We added a reference to this (for anonymity yet without the true url) to the introduction. We are unsure how a simulation can make it even easier to reproduce and compare to our results, but we are happy to take further suggestions.
>
> > Similarly, the experiments in Section 4.3 need to be empirically more rigorous. What happens after 250 secs? Why does the plot stop exactly when the segmentation quality peaks?
>
> The trajectory (and therefore the data recording) ends after 250secs, so there is nothing more happening. The peak could be an artifact, or point to a more substantial improvement after a longer time. Since we are unable to say which it is, we point out in the conclusion that longer-time deployments would be interesting to analyse. Unfortunately, we currently do not have longer recordings from our field experiments available.

---

> > ### Comment · Reviewer_jkw9 · 2021-08-25
> > **Reply to authors**
> >
> > _This will be published alongside the rosbag recordings... We are unsure how a simulation can make it even easier to reproduce and compare to our results, but we are happy to take further suggestions._
> > The rosbags are helpful, thank you. I understand that it's non-trivial to setup a simulated framework for benchmarking. But the original comment was more along the lines of: is the 10% improvement in localization specific to the environments presented in the paper? Or can you consistently show X% improvements in any given indoor environment with the proposed continual learning framework. The latter would obviously be a stronger result, but the rosbags partially addresses the concern because at least they can be replayed.

---

> ### Author Response · Authors · 2021-08-19
> **Answer to Reviewer jkw9: Part 1 of 2**
>
> > The self-supervised learning approach uses pseudo-labels generated by thresholding LiDAR scans to mask out foreground from background, but there is no baseline that investigates how good these pseudo-labels themselves are. What if you don't learn anything (self-supervised or continual) and directly use these pseudo-labels for segmentation? How well does this perform? The pseudo-labels might be noisy, but this depends on what sort of accuracy is necessary for the end-task, which is undefined. I am not sure if this pseudo-label result is already in Table 1; I had a hard time connecting the descriptions in Section 4.1 to items in Table 1.
>
> We investigate the segmentation accuracy of the pseudo-labels themselves in Appendix A.6. As we describe, this evaluation is strongly biased and therefore not presented in the main paper.
> It is of course not possible to use the pseudo-labels for the localisation, since the pseudo-labels are only available after successful localisation. However, it is possible to use a metric similar to the one we use for the pseudo-labels, i.e. we can remove points at a high distance from the floorplan directly within the LiDAR scans (without the additional visual information from our segmentation models). This is already what we do, even in the ‘no segmentation’ approach. So as the most basic baseline, we have a purely geometric method that filters out ‘outliers’ based on distance between LiDAR scan and floorplan. The exact filters that we apply to the scan are listed in Appendix A.5. We note that the parameters for this filtering (20% largest distances and filtering based on normals) are different from those used for the pseudo labels (0.1m distance). This is because these parameters are tuned towards their respective tasks/algorithms. Removing all points of >0.1m distance from the scan (at the initial guess before ICP) would lead to localisation failure in all environments.
>
> Concerning Table 1, we apologize for the confusion and hope that we could improve the manuscript. We compare 3 approaches:
>
> - ‘no segmentation’: All points of the LiDAR scan are considered valid and fed into the ICP registration. The first step in the registration is then to apply the filters from Appendix A.5.
> - ‘trained on NYU’: The LiDAR scan is filtered based on an image segmentation pretrained on an existing body of high-quality data (the NYU dataset [57]) and only points where the image is segmented as ‘foreground’ are fed into the ICP registration. The first step in the registration is then to apply the filters from Appendix A.5.
> - ‘self-improving’: same pipeline as ‘trained on NYU’, but the image segmentation network has been trained on images and pseudo-labels gathered from an independent trajectory in the same environment.
>
> We can therefore, given this data in Table 1, compare whether the image-segmentation based filtering is useful at all (it is, the localisation improves in general when using segmentation, even with the NYU-trained network). We can also measure the effectiveness of the training based on pseudo-labels: Both segmentation quality measured against annotated ground-truth images and localisation accuracy are better with the ‘self-improving’ model than training the network just on NYU.
> We hope referencing the methods in section 4.1 made the connection to Table 1 more clear. Please let us know if you would like to see further clarifications in the manuscript.
>
> > The results emphasize 60% gains in foreground-background segmentation from the proposed approach, but the paper does not mention a strong use-case for why foreground-background is necessary or important for a specific task. The current setup assumes access to LiDAR+vision and maps/models of the environment, so I am not sure what foreground-background segmentation offers in this scenario. Obstacle avoidance and localization can be done with LiDAR, without vision. Further, the segmentation labels do not classify object categories, so they cannot be used for task-planning.
>
> Target applications for localisation in floorplans can be any indoor robots, since floorplans are available in abundance and remove any pre-mapping stages from deployment (see also our answer to reviewer Ncrf). For construction robots in particular, localisation with respect to a floorplan is required, since construction tasks are mostly defined in such floorplans or 3D building models.
> However, registration between pure LiDAR scans and floorplans is difficult, and can lead to failures such as the one shown in Table 1. Table 1 therefore clearly shows the advantages of the additional camera-based background-foreground-segmentation for localisation in floorplans.
> We extended the motivation in the introduction, please let us know if there is still something missing.

---

> > ### Comment · Reviewer_jkw9 · 2021-08-25
> > **Reply to authors.**
> >
> > Thanks for the detailed response. Thanks for the pointers to pseudo-label results in the Appendix.
> >
> > I am still a bit confused on why the pseudo-labels themselves cannot be used to improve localization. You can run the localization with 'no segmentation', get pseudo-labels, and then use these labels to mask out foreground-background to improve the localization next timestep. I understand that the pseudo-labels are supposed to be used for continual learning, but this is a basic sanity check (from a scientific standpoint) to make sure that the core problem cannot be solved without _any_ learning at all.
> >
> > _We investigate the segmentation accuracy of the pseudo-labels themselves in Appendix A.6. As we describe, this evaluation is strongly biased and therefore not presented in the main paper._
> > This is a bit worrying. As stated in the original review and the comment above, this is an important reference point for all results.
> >
> > _L583: Unfortunately, we could not recover pseudolabels for the images that were used to generate ground-truth in the office environment._
> > I am not sure what happend here.

---

> > > ### Author Response · Authors · 2021-08-30
> > > **Answer to Reviewer jkw9**
> > >
> > > We thank the reviewer for engaging in the discussion and going through our comments.
> > >
> > > The suggested "double localisation" seems doable. Unfortunately, all other reviewers already requested additional experiments earlier (ablation of the full method over different ICP parameters, more online-learning runs, and deployment into 4 subsequent environments), which made it unfeasible to generate results in time. It would of course not be a realistic method, given already the 1.3s for pseudolabel generation (see Appendix A.1). We note that it would take just a single registration that is off by more than 10cm for the pseudolabels to label the whole scene as foreground. This would be just a minor noise in learning, but cause failure when used for localisation directly.
> > > In general, we would like to reiterate that our core contribution is the proposal of combining self-supervision and continual learning. Our experiments test this framework on the problem of localisation in floorplans to investigate the effectiveness of the proposed system, and we never claim that different solutions to the application would not be possible. However, our newly added ablation study in Appendix A.5 suggests that also with increased baseline accuracy, our system yields improvements.
> > >
> > > > L583: Unfortunately, we could not recover pseudolabels for the images that were used to generate ground-truth in the office environment.
> > > I am not sure what happend here
> > >
> > > There was a problem with the time-synchronisation for the recording that we used to annotate ground-truth. This means that we could not recover the corresponding LiDAR scan for the frames of the ground-truth annotation. We only discovered this after the annotation was completed. However, we made an effort to manually align three frames in Figure 9 to allow for a qualitative comparison. Figure 9 also illustrates the bias when evaluating the pseudo-labels quantitatively, given the large ignored parts.

---

### Official Review · Reviewer_XC6X · 2021-07-23

**Originality:** Good
**Technical Quality:** Good
**Clarity Of Presentation:** Fair
**Impact:** 3

**Recommendation:**

Weak Accept: I recommend accepting the paper, but will not argue for my recommendation if the majority of other reviewers have a different opinion.

**Summary:**

This paper proposed a self-supervised continual learning strategy on scene perception where the perception task is a background/foreground binary classification. Self-supervised learning is based on generated pseudo-labels on localised LiDAR scan, where each point are labelled based on how close to the nearest floorplan via a defined threshold. By further applying a memory buffer storing on 10% of previous scene data, it achieves continual learning and shows that training on new scenes could help learn a better representation improving performances of the previous scenes.

**Issues:**

See weaknesses.

**Reviewer Expertise:**

Good: General knowledge of the area

**Strengths And Weaknesses:**

The general concept of this paper is appealing, and I appreciate the video attached shows how this robot with the proposed method performs in a real-world environment.

Some additional comments:

-- Limited novelty: The proposed method is based on components that are standard and commonly used in the community, for example, the random buffer is typically used in perception-based continual-learning systems, and in most RL-based methods. Though I agree the design of pseudo-labeling is new and interesting, but limited to the task of only performing binary segmentation is a bit too easy. The background class of the indoor room has a very clear geometric structure and thus transferring across different tasks would be simple and straightforward. Considering background and foreground segmentation as "perception" seems to be an over-claim to me.

-- Quality of visualisation: In the video, It's very hard for me to identify the ground-truth tracking and predicted tracking.

-- Compared to other CL baselines: To the best of my knowledge (but happy to be corrected if wrong), EWC and other presented regularisation-based CL methods are designed with the assumption that we don't have access to any of the data in the previous tasks. While in the replay-based methods, the data are stored in the memory buffer and thus would be still trained when deploying in the new scenes. So it's a bit unfair to directly compare to these regularisation methods without properly justifying the differences.

=======================================================

The authors have clarified all my questions in the rebuttal phase. And I decide to raise my final rating to Weak Accept.

**Summary Of Recommendation:**

I like the overall system to be shown and work in a real-world environment. However, the paper is a bit overclaim on the impact and the novelty of the method.

---

> ### Author Response · Authors · 2021-08-19
> **Answer to Reviewer XC6X**
>
> > Limited novelty
>
> The core novelty of our paper is to investigate the combination of continual learning and self-supervision on a robotic system, i.e. we use mostly existing building blocks in a novel combination. This combination of self-supervision and continual learning has in our opinion a high potential for many robotic applications, because:
>
> - it enables learning in a fully autonomous way,
> - it has the potential to solve many issues around the open-world problems that affect currently used approaches that train on fixed datasets [A, B],
> - and we specifically discuss in Section 4.5 how other self-supervision approaches can be incorporated to extend our approach to more classes and applications.
>
> We are not aware of any prior works that investigate this combination, but are happy to discuss suggestions.
> Finally, it is true that we do not develop a new method to prevent forgetting, but it would also be a big stretch to say that it was clear prior to our work which method would perform well on real-world robotic data, or in a self-supervised setting.
>
> > The background class of the indoor room has a very clear geometric structure and thus transferring across different tasks would be simple and straightforward.
>
> To avoid any confusion: Our segmentation is based on images, the networks do not have any access to LiDAR data. Visually, as can be inspected in the examples from page 16 onwards, the wall texture is very different between all environments. Please let us know if we did not understand this comment correctly or if you are still under the impression that the domain gap between the tested environments is not sufficiently large.
>
> > So it's a bit unfair to directly compare to these regularisation methods without properly justifying the differences.
>
> Thank you for pointing this out. We agree that a discussion of the drawbacks of memory replay was so far missing. If we understand the reviewer correctly, the reviewer is worried that readers that are unfamiliar with continual learning literature may not understand why the regularisation based methods do not perform equally well. We added an explanation to section 4.4. Please let us know if we understood this comment correctly and if you are satisfied with our explanation in section 4.4.
>
> > Considering background and foreground segmentation as "perception" seems to be an over-claim to me.
>
> We certainly do not want to claim anything that we do not contribute. While foreground-background-segmentation is an important perceptual task for humans [C], we agree that foreground-background-segmentation in images alone is not yet perception. However, the proposed perception system encompasses multiple sensors and leverages different parts (localisation + vision) of a whole robotic system. An important part of our work is to show how improvements in foreground-background-segmentation also improve localisation, and we therefore found it reasonable to call this “improving perception”. We are however happy for suggestions on what could be a more accurate term in your opinion.
>
> [A] Sünderhauf, N., Brock, O., Scheirer, W., Hadsell, R., Fox, D., Leitner, J., Upcroft, B., Abbeel, P., Burgard, W., Milford, M., & Corke, P. (2018). The limits and potentials of deep learning for robotics. The International Journal of Robotics Research, 37(4-5), 405–420. https://doi.org/10.1177/0278364918770733
> [B] Zendel, O., Honauer, K., Murschitz, M., Steininger, D., & Fernandez Dominguez, G. (2018). Wilddash-creating hazard-aware benchmarks. Proceedings of the European Conference on Computer Vision (ECCV), 402–416. https://doi.org/10.1007/978-3-030-01231-1_25
> [C] Mazza, V., Turatto, M., & Umilta, C. (2005). Foreground–background segmentation and attention: A change blindness study. Psychological research, 69(3), 201-210. https://link.springer.com/content/pdf/10.1007/s00426-004-0174-9.pdf

---

> > ### Comment · Reviewer_XC6X · 2021-08-23
> > **Reply to Authors**
> >
> > I'd like to thank the authors' detailed comments and clarifications. The new paper now definitely looks better.
> >
> > Specifically, considering the CL baseline, I would suggest just keeping one (maybe EWC) regularisation method, and leaving some space to evaluate "self-improving" more than 2 scenes. At the current stage, all evaluations were done in a source-to-target type of learning. But in order to really show "self-improving" in a more general sense, I would suggest to do all 4 datasets NYU, garage, construction, office, with two or three different orders.
> >
> > That can show two things:
> >
> > 1. The generalisation of the proposed CL method with the scale of more number of scenes, hopefully, robust to the choice of orders.
> > 2. How learning with more different numbers of scenes would change the segmentation quality of previous scenes.
> >
> > If the authors can show these experiments, I would happy to raise my scores to acceptance.

---

> > > ### Author Response · Authors · 2021-08-30
> > > **Evaluation on 4 Subsequent Environments Completed**
> > >
> > > We are happy to inform the reviewer that we were able to complete the suggested experiments and adapted Section 4.2 accordingly.
> > >
> > > We found space to incorporate the result of two inverse runs (to show different orders as suggested) into the main paper in Table 2, and present all other orders in Tables 5 and 6. We find that our system scales well to the increased number of scenes, regardless of their order. Similar to the other results in Table 2, we also find cases where deployment into more environments improves over the single environment deployment, suggesting positive backward and forward transfer.

---

> > > > ### Comment · Reviewer_XC6X · 2021-08-31
> > > > **Final Response**
> > > >
> > > > Thanks for the effort. All my concerns and questions have been clarified.
> > > >
> > > > And I will take weak acceptance as my final rating.

---

### Official Review · Reviewer_bspZ · 2021-07-31

**Originality:** Good
**Technical Quality:** Very Good
**Clarity Of Presentation:** Very Good
**Impact:** 3

**Recommendation:**

Weak Accept: I recommend accepting the paper, but will not argue for my recommendation if the majority of other reviewers have a different opinion.

**Summary:**

The authors present a self-improving perception framework in the context of robot localization against 3D models/floor plans of changing environments. The localization is aided by a DNN, which segments the current scene into foreground (novel objects, not used for scan matching) and background (consistent with the 3D model, used for scan matching) points. The focus of this work is on how to train this network in a self-improving manner while adapting to new environments. To this end, authors propose a way to generate noisy pseudo labels from the current robot pose estimate, the current scan and the available 3D model. These pseudo labels are then used to train the DNN and thus improve its performance in the new environment. In turn, the better segmentation improves the localization itself. To avoid forgetting of already acquired knowledge during the deployment in other environments, authors use continual learning techniques such as memory replay. The experiments on a real robotic platform demonstrate that the approach is able to improve both localization and segmentation in novel environments, transfer acquired knowledge across environments and avoid catastrophic forgetting.


**Issues:**

* For a less informed reader (like myself) on continual learning, could you please expand on the significance of the finding that CL works with noisy pseudo labels? Given the nature of memory replay and that noisy pseudo labels have been successfully used in self-training settings, I don’t find this as a surprising or very insightful outcome. This is a genuine question, without trying to reduce your contribution.
* For the localization, do you use robot’s odometry?
* How would your approach scale in terms of accuracy, forgetting and computational efficiency when the robot is deployed in more than two environments e.g. garage -> construction -> office, etc.?
* How does your work compare to *Self-Supervised Learning of Lidar Segmentation
for Autonomous Indoor Navigation, Hugues et al., ICRA'21*? I understand that it is a very recent and parallel work, but I think it should be discussed in the related work.
* The online learning experiment (Sec. 4.3) seems to be inconclusive and the description lacks details e.g. in which environment the experiment is performed? Is the network initially pre-trained on the NYU dataset? Do you observe similar outcomes in the rest of the environments?
* In general, for most of the experiments one could think of applying several iterations (like it is usual in self-training) i.e. improved segmentation -> improved localization -> improved pseudo labels -> improved segmentation -> ... Have you tried that?


**Reviewer Expertise:**

Fair: Some knowledge of the area

**Strengths And Weaknesses:**

Robots are deployed in a large variety of environments and the ability to adapt to new domains is a sought-after skill. Therefore, the paper contributes to a very relevant topic for the robotics community. It is well written and contains comprehensive experiments with valuable insights. I especially appreciate that the experiments were conducted on a real-world robotic platform.

While the self-improving and continual learning framework is interesting, I find its demonstration/application scenario to be less appealing. Potentially, it is because the importance of the removal of the foreground points for the task at hand is not stressed enough in the introduction (unfortunately, the related citation is anonymized). I think one or two sentences elaborating on this matter could be beneficial for the reader. The chosen localization baseline with point-to-plane ICP seems to be quite basic and I was left wondering if with more robust scan-matching (Beyond points: Evaluating recent 3D scan-matching algorithms, Magnusson et al., ICRA'15) or a SLAM method (with e.g. lidar odometry which could even leverage the existence of new objects) would perform better at the localization task and in turn in foreground/background segmentation (with the same approach used for the generation of pseudo labels). This is neither discussed in the text nor investigated in the experiments. Also, if the authors could suggest more applications of the learnt binary segmentation, it could further highlight the importance of this work.

**Summary Of Recommendation:**

I simply think that the topic addressed in this work is very relevant to the robotics community and this work provides some valuable contributions.

---

> ### Author Response · Authors · 2021-08-19
> **Answer to Reviewer bspZ: Part 1 of 2**
>
> > [...] the importance of the removal of the foreground points for the task at hand is not stressed enough in the introduction (unfortunately, the related citation is anonymized). I think one or two sentences elaborating on this matter could be beneficial for the reader.
>
> Thank you for pointing this out. We updated the introduction to better explain the application scenarios for localization in floorplans. Please let us know if you find the application now sufficiently motivated.
>
> > The chosen localization baseline with point-to-plane ICP seems to be quite basic and I was left wondering if with more robust scan-matching (Beyond points: Evaluating recent 3D scan-matching algorithms, Magnusson et al., ICRA'15) or a SLAM method (with e.g. lidar odometry which could even leverage the existence of new objects) would perform better at the localization task and in turn in foreground/background segmentation (with the same approach used for the generation of pseudo labels). This is neither discussed in the text nor investigated in the experiments.
>
> We thank the reviewer for this important question. First of all, we would like to note that SLAM methods are not suitable in our context, because we investigate the localisation in a given map (in the form of a floorplan).
> The referenced paper concludes that NDT is preferable in situations with small overlap or poor initial alignment, while we assume a good initialisation and have a large overlap between scan and map. In their presentation slides (http://130.243.105.49/Research/Learning/publications/2015/Magnusson_etal_2015-ICRA15-Beyond_Points-spotlight.pdf), the authors actually recommend “Use point2plane ICP for data with large overlap in corridors”. Furthermore, we do not use the most basic implementation of point2plane ICP, but apply a number of (commonly used) processing steps to increase robustness. This is described in lines 128-129 and listed in Appendix A.5.
> In general, we consider the choice of the registration technique an orthogonal research question to our work. Our pseudo-label generation is independent of the registration method and simply requires a pose estimate, while also the segmentation only filters the input that can be registered with any algorithm. We understand the concern that this choice of algorithm can be clarified in the text (where so far we were only referencing a prior work of ours), which is why we added a footnote to Section 3.1.
>
> > For a less informed reader (like myself) on continual learning, could you please expand on the significance of the finding that CL works with noisy pseudo labels? Given the nature of memory replay and that noisy pseudo labels have been successfully used in self-training settings, I don’t find this as a surprising or very insightful outcome. This is a genuine question, without trying to reduce your contribution.
>
> The problem with pseudo-labels like ours in comparison with training on high-quality labels is that the optimized loss does not reflect the true training objective. We train on pseudo-labels but actually want to increase performance on the ground-truth test data. Of course, the advantage in our scenario is that the pseudo-labels are from the deployment environment, but the more noisy these pseudo-labels are, the more difficult is this objective. In memory replay, we increase the difficulty even further since now we have a multi-objective training (performance of past and present tasks), without representing the true past task (we only have pseudo labels for that one) or the true current task (also only pseudo labels). It was therefore indeed a surprising outcome for us how well this worked, in particular when compared to the distillation approach which in previous works performed well with high-quality labels [38].
> We are not entirely sure what the reviewer means with “self-training”, but we assume that self-training actually means the same as “output distillation” from Section 4.4.
>
> > For the localization, do you use robot’s odometry?
>
> No, in these experiments, the localization is purely based on registration as described in Section 3.1.
>
> > How would your approach scale in terms of accuracy, forgetting and computational efficiency when the robot is deployed in more than two environments e.g. garage -> construction -> office, etc.?
>
> For three environments, we present Table 5 in the Appendix that shows the progression of forgetting when moving from NYU -> Scene A -> Scene B. We still got satisfying results with the 2 replay buffers. However, as also the added discussion to Section 4.4 notes, this does not scale infinitely. Practically, we would expect that for many applications, concurrent improvement in 3 environments is sufficient. However, research progress on regularisation based methods hopefully enables higher scalability in the future.

---

> ### Author Response · Authors · 2021-08-19
> **Answer to Reviewer bspZ: Part 2 of 2**
>
> > How does your work compare to Self-Supervised Learning of Lidar Segmentation for Autonomous Indoor Navigation, Hugues et al., ICRA'21? I understand that it is a very recent and parallel work, but I think it should be discussed in the related work.
>
> Thanks for pointing us to this interesting work. Compared to our self-supervision component, this is indeed a parallel and similar work for a slightly different application. As we discussed in Section 4.5, we expect the proposed framework to be compatible with many different self-supervision approaches. We would expect that our system could easily incorporate this self-supervision approach, assuming that their results from simulation transfer well to real-world deployment. We added it therefore to the related work and referenced it in Section 4.5.
> The big difference is of course that Hugues et al. do not consider the continual learning aspect of the improvement, and do not evaluate or mitigate forgetting.
>
> > The online learning experiment (Sec. 4.3) seems to be inconclusive and the description lacks details e.g. in which environment the experiment is performed? Is the network initially pre-trained on the NYU dataset? Do you observe similar outcomes in the rest of the environments?
>
> The online experiment was conducted in the construction environment, as noted in the caption of Figure 4 (previously figure 5). Yes, the model is again pretrained on NYU, thanks for pointing this out, we added it to the text.
> We agree that a more thorough investigation of which frames to keep in the training data buffer, which learning rate to apply, at what speed to train etc. would be interesting for the online learning, but would by far exceed the page limit for this work. Instead, we added this (small) experiment to demonstrate that this capability exists at all, because we consider it very useful and exciting for future work.
> Given the space limitations, we can offer to try adding an online learning run for a different environment to the Appendix in the next week. Please let us know.
>
> > In general, for most of the experiments one could think of applying several iterations (like it is usual in self-training) i.e. improved segmentation -> improved localization -> improved pseudo labels -> improved segmentation -> ... Have you tried that?
>
> This is an interesting suggestion. On the presented system in this paper, we did not try such iterative improvement for larger iterations. The online learning of course is an edge case of such a setting for iteration-size of 1 batch.

---

> ### Author Response · Authors · 2021-08-26
> **Update for Reviewer bspZ**
>
> We would like to inform you that we just uploaded a revised Appendix that includes online learning experiments in the two other environments. The third row (evaluation of snapshots from different timesteps on a reference trajectory) is very time-consuming, and other reviewers also asked for some additional evaluations.
> We observe similar trends as Figure 4 in all environments. Similar to Table 1, the improvements in segmentation and localisation are not always proportional.
> Please let us know if you have further questions about this.

---

> ### Comment · Reviewer_bspZ · 2021-09-03
> **Final recommendation**
>
> Dear Authors,
>
> thank you for your clarifications and the efforts in improving the paper. Most of my concerns have been addressed. I will stay with my original recommendation of weak accept.

---

### Meta-Review · Area_Chair_qUHT · 2021-08-13

**Recommendation:** Accept (Poster)
**Confidence:** 5

**Metareview:**

The paper proposes a continual learning perception framework for indoor robot localization on floor plans. All the reviewers find the problem being addressed in this work of importance and a required skill for robots. The paper is well written and clearly organized with real-world robotic experiments that are appreciated by the reviewers. Most of the concerns raised by the reviewers have been thoroughly addressed in the rebuttal. I thank the authors for the engaging discussions during the rebuttal. Some minor concerns still exist, primarily including a  pseudo-label baseline. Nevertheless, I agree with the reviewers that the paper is definitely an interesting contribution.

---

### Decision · Program_Chairs · 2021-09-13

**Decision:**

Accept (Poster)

**Comment:**

The paper proposes a continual learning perception framework for indoor robot localization on floor plans. All the reviewers find the problem being addressed in this work of importance and a required skill for robots. The paper is well written and clearly organized with real-world robotic experiments that are appreciated by the reviewers. Most of the concerns raised by the reviewers have been thoroughly addressed in the rebuttal. I thank the authors for the engaging discussions during the rebuttal. Some minor concerns still exist, primarily including a  pseudo-label baseline. Nevertheless, I agree with the reviewers that the paper is definitely an interesting contribution.